# Primary Cardiac Intimal Sarcoma Visualized on 2-[^18^F]FDG PET/CT

**DOI:** 10.3390/diagnostics10090718

**Published:** 2020-09-18

**Authors:** Kim Francis Andersen, Nahid Sharghi Someh, Annika Loft, Jane Maestri Brittain

**Affiliations:** Department of Clinical Physiology, Nuclear Medicine & PET, Rigshospitalet, Copenhagen University Hospital, Blegdamsvej 9, DK-2100 Copenhagen, Denmark; nahid.sharghi.someh@regionh.dk (N.S.S.); annika.loft.jakobsen@regionh.dk (A.L.); jmbrittain@dadlnet.dk (J.M.B.)

**Keywords:** primary cardiac tumors, malignancy, cardiac intimal sarcoma, 2-[^18^F]FDG PET/CT

## Abstract

Primary cardiac tumors are extremely rare, with an incidence of 0.001–0.03%. Twenty-five percent of these tumors are malignant, with sarcomas accounting for approximately 95%. Cardiac intimal sarcoma is the least reported subtype of primary cardiac sarcoma. These endocardial mesenchymal tumors most often arise from great arterial vessels, and are rarely located in the heart. They often present with an aggressive clinical course and have a poor prognosis, with surgical resection with achievement of free margins being the mainstay of treatment. This emphasizes the importance of an early, correct diagnosis and timely intervention. We report a 60-year-old Caucasian male with several former cardiac surgical procedures due to congenital aortic stenosis, presenting with functional mitral stenosis/insufficiency and left ventricular outflow tract obstruction (LVOTO) due to massive masses in the left ventricle and atrium of the heart. Hybrid imaging with 2-deoxy-2-[^18^F]fluoro-D-glucose positron emission tomography/computed tomography (2-[^18^F]FDG PET/CT) was performed prior to surgery to characterize the intracardiac masses and estimate tumor burden, as well as to identify a potential extracardiac primary malignancy.

**Figure 1 diagnostics-10-00718-f001:**
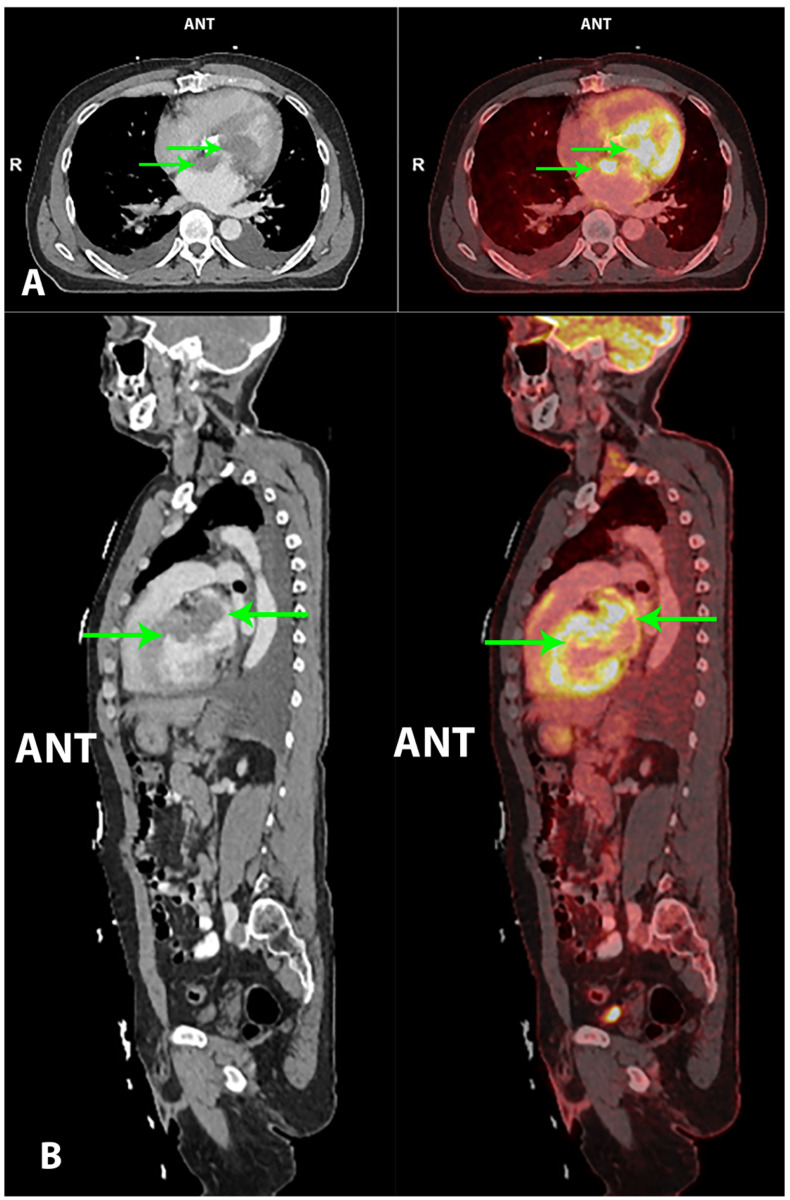
A 60-year-old Caucasian male with several former cardiac surgical procedures due to congenital aortic stenosis—in chronological order: (1) aortoplasty; (2) mechanical aortic valve replacement; (3) aortic prosthetic graft. As a consequence, the patient was treated with angiotensin II receptor blockers and anticoagulants. Prior to admission, the patient had experienced night sweats and malaise for one month, but no weight loss. There were complaints of progressive dyspnea the last week upon admission. The patient had no tendency to infections. On admission, the patient was hemodynamic stable and nonfebrile. Initial laboratory data came out with an elevated C-reactive protein (CRP; 129 mg/L) and lactate dehydrogenase (LDH; 263 U/L), and an international normalized ratio (INR) of 2.5. First-line imaging at the primary hospital with transesophageal echocardiography (TEE) demonstrated a massive thrombus/tumor mass adjacent to the mitral valve expanding into the left ventricular outflow tract, with physiology consistent with mitral stenosis/insufficiency and left ventricular outflow tract obstruction (LVOTO). The patient was transferred to a secondary hospital, where a cardiac CT showed massive masses in the left atrium in relation to the atrial septum and the mitral valve, and also in the left ventricle in relation to the mechanical aortic valve. The masses demonstrated low density (~50 Hounsfield units), suggestive of a thrombus or a tumor with very low vascularity. There were also several enlarged lymph nodes in the mediastinum. Consequently, the patient was transferred to a tertiary hospital with a specialized cardiothoracic surgical department, due to potential resection of the intracardiac thrombus/tumor masses. Hybrid imaging with 2-deoxy-2-[^18^F]fluoro-D-glucose positron emission tomography/computed tomography (2-[^18^F]FDG PET/CT) was performed pre-operatively, to characterize the intracardiac masses and estimate tumor burden, as well as to identify a potential extracardiac primary malignancy, which could alter the planned surgical procedure. Imaging performed 1 h post-injection of 4.0 MBq/kg 2-[^18^F]FDG showed highly metabolically active intracardiac masses in the left atrium and ventricle, with suspiciousness of malignancy (maximum standardized uptake value (SUV_max_) of 21.2)—marked with green arrows on both the CT (left) and fused PET/CT (right) transaxial Figure 1 (**A**) and sagittal Figure 1 (**B**) images. There were no extracardiac malignant findings, i.e., the aforementioned mediastinal lymph nodes were interpreted as benign. Tumorectomy in toto was performed with the reconstruction of the heart with patches in the left atrium and ventricle, as well as biological aorta/mitral valve replacement. The immediate postoperative course was complicated with lung edema, fever, liver dysfunction and cardiac arrhythmia. Based on the tumor’s morphology, location, fluorescence in situ hybridization (FISH), next generation sequencing (NGS) and immunohistochemical profile, final pathology was cardiac intimal sarcoma. Unfortunately, the surgical resection margin was not tumor-free. The patient’s case was discussed at a multidisciplinary tumor board. Oncological treatment options were severely limited, and further surgical treatment was not an option in this case. The patient was transferred to his local hospital and follow-up was planned according to national guidelines. Unfortunately, the patient’s clinical condition worsened rapidly, and he died two months later. The patient died during the illness, and was thus not able to give consent. He is not identifiable in the present case, and relatives were not available for consent. The approval for the use and publication of patient data was given by The Danish Data Protection Agency in the Capital Region of Denmark (approval #: P-2020-883, approved on 11-09-20).

This case illustrates the aggressive nature of a rare disease. Based on autopsy findings, primary cardiac tumors are extremely rare, with an incidence of 0.001–0.03%. Overall, 25% of these tumors are malignant, with sarcomas accounting for approximately 95% [1,2]. Even though a retrospective analysis of 100 primary cardiac sarcomas showed that intimal sarcoma was the most frequent [3], this subtype remains the least reported subtype of primary cardiac sarcoma [1,4]. These endocardial mesenchymal tumors most often arise from great arterial vessels, and are rarely located in the heart [3]. Even though patients may be asymptomatic for a long time, most signs and symptoms of these tumors result from intracardiac obstruction, dependent on the location, size, growth rate, invasiveness and friability of the tumor. More nonspecific symptoms in terms of weight loss, fever and fatigue may also be present [5].

The optimal imaging diagnostic work-up of suspected cardiac masses is not well established. However, both echocardiography (especially TEE) and magnetic resonance imaging (MRI) is considered to represent sensitive techniques to detect and characterize these lesions. As in our case, TEE often is used as first-line imaging modality and may demonstrate the cardiac tumor, its extent and hemodynamic effects. Cardiac MRI may provide superior information in terms of the location, morphology and extent of the mass, and as such, may be considered as the reference noninvasive imaging technique [6]. Exactly why cardiac CT was preferred to cardiac MRI as imaging modality at the secondary hospital in this specific case is unclear. However, cardiac CT can provide useful anatomic and functional information as an adjunct to echocardiography and MRI in the evaluation of cardiac lesions, and can serve as an alternative to MRI, due to high contrast and spatial resolution, fast acquisition times, and the capability to identify calcification and fat. Moreover, cardiac CT may have specific advantages in defining the cardiovascular extent of the mass and excluding coronary artery disease prior to possible surgery [7].

In our case, at the time of the performed 2-[^18^F]FDG PET/CT, there was still a wide differential diagnosis, ranging from a benign thrombus or benign primary cardiac tumor to a malignant mass either in terms of a malignant primary cardiac tumor or cardiac metastases with unknown primary tumor. However, findings were compatible with a malignant primary cardiac tumor with no signs of peri-/extracardial metastatic disease. Consequently, curative intended surgical removal of the tumor was performed. Moreover, 2-[^18^F]FDG PET is not yet included in the routine diagnosis of cardiac masses, which is probably due to both the low incidence and to the often seen physiologically high uptake of 2-[^18^F]FDG in the myocardium [6]. Although not used in our case, the latter may be reduced by combining a low-carbohydrate diet with prolonged fasting prior to injection of the radiotracer, resulting in a shift in myocardium metabolism towards fatty acid consumption, and thereby an enhanced lesion to background 2-[^18^F]FDG uptake ratio. This technique may be extremely useful in the characterization of cardiac tumors. Even though the level of evidence is very low, several reports demonstrate the usefulness of hybrid imaging with 2-[^18^F]FDG PET/CT or 2-[^18^F]FDG PET/MRI in the discrimination between benign and malignant cardiac masses based on the quantification of tracer uptake—it should be kept in mind that discriminating a benign thrombus from a malignant cardiac lesion may be challenging, as both lesions can demonstrate an increased 2-[^18^F]FDG uptake. However, studies performed by Nensa et al. [8] and Rahbar et al. [9], respectively, both demonstrated a significantly higher SUV_max_, both in malignant primary and secondary cardiac tumors compared to non-malignant cardiac lesions, with the latter study assessing a SUV_max_ cutoff with high sensitivity at 3.5. This was in concordance with the reported findings suggestive of a malignant primary cardiac tumor in our case. Additionally, these imaging modalities may be useful in the setting of: (1) estimation of disease stage and tumor burden; (2) follow-up with evaluation of residual disease after surgical removal of tumor; (3) early response evaluation of neoadjuvant and adjuvant therapy [6,8,9].

As in our case, cardiac intimal sarcomas often present with an aggressive clinical course, and have a very poor prognosis. Surgical resection with achievement of pathologically negative margins (R0 resection) is the mainstay of treatment, associated with prolonged survival. However, local recurrence and metastatic disease often occur within the first year, despite complete surgical resection of the tumor [10,11]. Oncological neoadjuvant and adjuvant treatment options are limited, clinical evidence is sparse and conflicting reports exist. Application of radiotherapy is associated with myocardial radiation injury and a substantial risk of chronic pericarditis and cardiomyopathy. Neoadjuvant chemotherapy may be used to reduce tumor burden in order to facilitate the complete surgical removal of the cardiac tumor. In the adjuvant setting, no standardized evidence-based chemotherapy regimen currently exists. Some studies have reported improved survival for patients treated with post-operative adjuvant chemotherapy [10,12], while others failed to demonstrate any modification of the clinical course of the disease [13,14]. Surgical debulking in a palliative setting may be performed in patients with rapidly progressing symptomatic disease. Palliative chemotherapy may be temporary beneficial when the primary tumor is not resectable, or in the presence of metastatic disease [6].

To summarize, the rareness and severe complexity of primary cardiac sarcomas pose serious challenges and require a multidisciplinary, highly specialized approach throughout the diagnostic workout, treatment and follow-up. Even so, the prognosis for these patients remains dismal.

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
