# Peer review of "Primary Cardiac Intimal Sarcoma Visualized on 2-[18F]FDG PET/CT"

_diagnostics, 2020, doi:10.3390/diagnostics10090718_

Round 1

Reviewer 1 Report

the authors report one case of a patient with intimal cardiac sarcoma detected by FDG-PET. The condition is rare, and usually PET scan is not required because of better performances of other imaging modalities in this setting. In particular transesophageal echocardiogram and magnetic resonance imaging provide anatomical details useful to distinguish masses from thrombi. Both these imaging procedures are not described and performed in the patient of actual case. The lack of a comparison with them represents the main drawback of the report. This appears relevant considering the possibility of FDG uptake in benign and malignant thrombi.

Author Response

Dear reviewer

We are grateful for your useful comments. Our reply as follows:

  • The authors report one case of a patient with intimal cardiac sarcoma detected by FDG-PET. The condition is rare, and usually PET scan is not required because of better performances of other imaging modalities in this setting. In particular transesophageal echocardiogram and magnetic resonance imaging provide anatomical details useful to distinguish masses from thrombi. Both these imaging procedures are not described and performed in the patient of actual case. The lack of a comparison with them represents the main drawback of the report. This appears relevant considering the possibility of FDG uptake in benign and malignant thrombi.

Reply: As described in line 70-75, the authors advocate for the application of TEE and cardiac MRI as sensitive techniques in terms of detecting and characterizing cardiac masses. It is not correct that the patient did not have an TEE; please see line 35-36 where be describe the findings on TEE. Instead of a cardiac MRI, the clinicians choose to perform a cardiac CT. The findings on this imaging modality are described in line 37-40. Our main purpose with this case report, though, is to demonstrate settings where hybrid maging with FDG-PET/CT or FDG-PET/MRI may be useful.

However, we fully agree regarding the possible difficulties in terms of discriminating a benign thrombus from a malignant cardiac tumor, as both lesions may show an increased FDG uptake. We have added our thoughts regarding this dilemma to the manuscript (line 90-91, marked with yellow).

Reviewer 2 Report

Diagnostics-860436

This is an interesting case report showing the FDGPET/CT imaging of rare cardiac intimal sarcoma. The images showed cardiac tumor nicely, and the presentation was well described. This report seems to be valuable, while I would like to recommend some revision.

  1. Title and the text:

2-[18F]FDG PET/CT is not the standard abbreviation, [18F]FDG PET/CT may be more popular. Note; Labeling with fluorine-18 is important, but not the labelled position of 2 because no other choice.

  1. line 81-83.

Several reports indicate that a low-carbohydrate diet, associated with prolonged fasting, is adequate for shifting myocardium metabolism toward fatty acid consumption, resulted in lowering the myocardial physiological FDG uptake and enhancing the lesion to background contrast. This has already been described in guidelines and review articles, and extremely useful not only for cardiac sarcoidosis but also for cardiac tumor. It was described in ref [6] in your list of references. Even though you have not used this technique to this patient, please add some discussion on this point.

Author Response

Dear reviewer

We are grateful for your useful comments. Our reply as follows:

  • Title and the text: 2-[18F]FDG PET/CT is not the standard abbreviation, [18F]FDG PET/CT may be more popular. Note; Labeling with fluorine-18 is important, but not the labelled position of 2 because no other choice.

Reply: Spelling radiotracers correctly may seem like a science in itself. However, according to the newly published EANM 'Guidance to Radiotracer Nomenclature' (https://www.eanm.org/content-eanm/uploads/2019/12/EANM_GUIDANCE-_TRACER_NOMENCLATURE-1.pdf), 2-[18F]FDG PET/CT is the recommended abbreviation, even though [18F]FDG PET/CT may be more popular and also can be used. Consequently, we have kept the abbreviation throughout the manuscript.

  • line 81-83: Several reports indicate that a low-carbohydrate diet, associated with prolonged fasting, is adequate for shifting myocardium metabolism toward fatty acid consumption, resulted in lowering the myocardial physiological FDG uptake and enhancing the lesion to background contrast. This has already been described in guidelines and review articles, and extremely useful not only for cardiac sarcoidosis but also for cardiac tumor. It was described in ref [6] in your list of references. Even though you have not used this technique to this patient, please add some discussion on this point.

Reply: We fully agree regarding the reviewer's comments about the usefulness of shifting myocardium metabolism towards fatty acid consumtion, and thereby enhancing the lesion to background FDG-uptake ratio. We have added our thoughts on this point in relation to the specific case (line 83-87, marked with yellow).

Round 2

Reviewer 1 Report

The response could be considered partially satisfying. The case description with the reference to the TEE (line 35-36), is not clear because the editing doesn't permit to differentiate clinical history form figure legend. Moreover the role of TEE must be emphasized  into the patient 's case because first level procedure. Finally the choice of CT instead of MRI require further clarifications.

The sentence in lines 89-91  should be elucidated  because may appear in contradiction; reference of role of PET in discriminating benign from malignant disease, as well as the cutoff values to do this,  must be reported at the end of the sentence.

Author Response

Dear reviewer/editor

We are grateful for the useful comments/suggestions made by the reviewer.

Our point-by-point resonse is as follows:

1) The case description with the reference to the TEE (line 35-36), is not clear because the editing doesn't permit to differentiate clinical history form figure legend. Moreover the role of TEE must be emphasized  into the patient 's case because first level procedure.

Response: Agreed. We have 'seperated' description of initial imaging in terms of TEE from the rest of the clinical story, added some findings on the TEE and emphasized the importance of the performed TEE into the patient's case.

2) Finally the choice of CT instead of MRI require further clarifications.

Response: Exactly why cardiac CT was preferred to cardiac MRI at the secondary hospital in this specific case is unclear. However, we have supplemented on possible explanations/advantages related to the application of cardiac CT in this specific setting. 

3) The sentence in lines 89-91  should be elucidated  because may appear in contradiction; reference of role of PET in discriminating benign from malignant disease, as well as the cutoff values to do this,  must be reported at the end of the sentence.

Response: We fully agree. Relevant references have been added. We have elucidated this part and related findings in our case to existing litterature.

On behalf of the co-authors and best regards,

Kim F. Andersen, MD